# Are Virtual Laboratories and Remote Laboratories Enhancing the Quality of Sustainability Education?

**Mark Ching-Pong Poo** [1,2,*] **, Yui-yip Lau** [3] **and Qiong Chen** [4]

1   Liverpool Hope Business School, Liverpool Hope University, Liverpool L16 9JD, UK
2   Liverpool Logistics, Offshore and Marine Research Institute, Liverpool John Moores University, Liverpool L3 5UX, UK
3   Division of Business and Hospitality Management, College of Professional and Continuing Education, The Hong Kong Polytechnic University, Hung Hom, Hong Kong
4   Navigation College, Jimei University, Xiamen 361021, China; qchen@jmu.edu.cn
*   Correspondence: pooc@hope.ac.uk

**Abstract:** Sustainability education, a multidisciplinary field demanding a profound understanding of intricate scientific, engineering, social and economic systems, necessitates innovative approaches. Laboratory experimentation plays a pivotal role in engineering and scientific education. The emergence of the COVID-19 pandemic heightened the importance of remote learning and home-based study in pedagogical practices. However, engineering education has faced challenges in adapting to novel teaching methodologies. A significant challenge during lockdowns was the effective delivery of laboratory experiences in virtual spaces. Virtual and remote laboratories, while not substituting the hands-on experience of physical labs, offered promising avenues to enhance learning during the disruption of in-person education. While most teaching activities transitioned seamlessly to online formats, laboratory sessions presented unique logistical challenges, including cancellations of fieldwork. Additionally, concerns arose regarding disparities in student achievement based on income levels. This study seeks to provide an overview of the implementation status of virtual and remote laboratories during the lockdown period in education. Its goal is to offer practical insights to improve the quality of learning experiences at home and in online settings.

**Keywords:** COVID-19 pandemic; remote laboratory; sustainability education; educational inequality

## 1. Introduction

Sustainability education is a multidisciplinary field that encompasses various disciplines, including environmental science, ecology, economics, engineering, sociology, anthropology, public health, and more [1]. It recognises the interdependence of environmental, social, and economic systems and seeks to create a more sustainable future by promoting responsible and ethical practices in all areas of human activity. Furthermore, it draws on the knowledge and expertise of multiple disciplines to provide students with a comprehensive understanding of sustainability challenges and opportunities. This multidisciplinary approach is necessary for addressing complex global challenges such as climate change, resource depletion, and social inequality, which require a holistic understanding of the interconnectedness of environmental, social, and economic systems.

Sustainability education is crucial for creating a more sustainable future by promoting responsible and ethical energy, water, food, and waste management practices [2]. It equips students with the knowledge and skills to address complex global challenges like climate change, resource depletion, and social inequality. These challenges require a multidisciplinary approach and a deep understanding of the interconnectedness of environmental, social, and economic systems. By providing students with a foundation in sustainability education, institutions are preparing them for the job market and ensuring that they have the skills and knowledge required to meet future workforce demands [3]. Sustainability is

becoming increasingly important in many sectors of the global economy, from renewable energy to sustainable agriculture.

Sustainability education is a dynamic field that encompasses a wide array of essential features. Sustainability education is described as "transformative education and requires transformational/sustainability pedagogies" [4]. Sustainability education is one of the 17 Sustainable Development Goals (SDGs). The role of sustainability education is to improve the skills and knowledge needed to contribute towards sustainable development. Sustainability education aligns with a sustainable curriculum created to reinforce students with interdisciplinary values and knowledge that are crucial to the pursuit of the concept of sustainability [5]. At its core, sustainability education is about instilling in students the critical thinking and problem-solving skills necessary to navigate the complex challenges our world faces in the realms of environment, society, and the economy. The adoption of sustainability education needs a rethinking of contents and objectives, a transformation of learning contexts within local networks and communities, and a redesign of curricula. Sustainability education can be further reinforced via innovation, education development, and educational practices [6]. Laboratory teaching activities serve as the practical application of theoretical knowledge, allowing students to bridge the gap between classroom learning and real-world problem-solving. These activities provide a hands-on, experiential approach to sustainability education, where students engage with tangible environmental and social issues. In the laboratory, they can apply their knowledge to analyse data, design experiments, and develop innovative solutions. This approach not only enhances their understanding of sustainability concepts but also equips them with the practical skills and multidisciplinary perspectives required to become effective agents of change. Through laboratory teaching activities, students are not only educated about sustainability, they become active participants in its realisation, making it a cornerstone of their academic journey and personal growth.

Sustainability education encourages students to develop the ability to analyse and interpret complex data and information, equipping them to assess the impact of their actions on the environment, society, and the economy [7]. It empowers them to make informed decisions grounded in evidence and rigorous analysis. Moreover, sustainability education promotes civic engagement by raising awareness of environmental and social issues and offering opportunities for students to participate in community-based sustainability initiatives [8]. It cultivates a sense of responsibility and commitment to the common good, encouraging active participation in creating a more sustainable future. This multifaceted approach cultivates a sense of responsibility and commitment to the common good, encouraging active participation in creating a more sustainable future [9]. Laboratory teaching activities are a pivotal component of sustainability education, providing students with practical, hands-on experiences to bridge the gap between theory and real-world problem-solving [10]. Students engage with tangible environmental and social issues through these activities, applying their knowledge to analyse data, design experiments, and develop innovative solutions. In doing so, they enhance their understanding of sustainability concepts and acquire the practical skills and multidisciplinary perspectives needed to become effective change agents [11].

Engineering and sustainability education are intrinsically intertwined. Engineers are critical in developing and implementing sustainable solutions to address pressing environmental and social challenges [12]. While engineering education equips students with the technical skills necessary for designing, building, and maintaining efficient and resilient systems, sustainability education broadens their perspective. It helps engineers understand the multifaceted impacts of their designs on the environment, society, and the economy, leading to the development of more sustainable solutions [13]. For instance, engineers well-versed in sustainability principles may design energy-efficient buildings, create eco-friendly transportation systems to reduce emissions, or pioneer renewable energy technologies.

Experiments are essential in the education of various areas as they can be used to improve the student's learning process. Whereas traditional hands-on labs offer students opportunities for experimentation with natural systems, they involve high costs associated with equipment, space, and maintenance staff [14]. During the pandemic, hands-on experiment opportunities have become rare, and most universities cannot provide not sufficient experimental sessions for students [15]. Therefore, this pandemic period can allow lecturers to implement more virtual [16] and remote [17] laboratories. However, they needed to be reviewed as they were rapidly substituted for the regular laboratory sessions.

Laboratory teaching activities are crucial to sustainability education because graduates require laboratory skills for their careers including the ability to carry out experiments, examine experimental data, and interpret the results [18]. As explained by Chiu [19], laboratory teaching activities play a vital role in sustainability education. One of the final goals of developing laboratory teaching activities is to improve perception, appropriate actions and cultivate responsible citizens for environmental sustainability. Indeed, laboratory teaching activities generate pedagogical implications for attaining educational goals. Developing the living learning laboratory has been identified as a promising pedagogy in sustainability education [20], supporting the Sustainable Development Goals (SDGs) recognised by the United Nations (UN) in 2015, notably quality education (SDG4) and reducing inequalities (SDG10) [18].

Virtual laboratories are based on software to simulate the lab environment, while remote lab, by definition, is an experiment that is conducted and controlled remotely through the internet. These experiments use real components or instrumentation away from the actual location from where they are being controlled or conducted [21]. Remote laboratories allow users to perform experiments and laboratory tasks over the internet without being near the equipment [22].

There are many common pros and cons to this new environment for laboratory training away from virtual laboratories [23]. On the right side, it can maximise time and space flexibilities to improve time efficiency and simplify complex procedures so that more complicated workshops can be carried out [24]. Furthermore, it offers a safe and convenient workshop environment for student learning assessments. Nevertheless, virtual workshops can also demotivate students from becoming familiar with physical instruments and real devices. Concerning the transferable skills such as teamwork and communication skills, which can often be found and delivered in traditional laboratory training, the remote access features in virtual and remote laboratory training could further discourage direct collaboration and interaction. Also, the risk of plagiarism in assessment is increased because of the online convenience platform (e.g., ChatGPT). As such, it may induce damage to honest students and professionals who make every effort to generate original products and content.

The adoption of virtual and remote laboratories has widely appeared in different educational levels ranging from primary schools to higher education, from technical colleges to self-education, and from vocational learning to university [25,26]. Similarly, their adoption discipline is expanding, namely, robotics, thermodynamics, control engineering, and spectroscopy, to name but a few. In doing so, the scientific paper publication and its research community are significantly emerging [27]. While virtual and remote laboratories have become more prevalent in e-learning, technology-enhanced learning, science education, and medical education, a significant research gap exists regarding their use in sustainability education [28]. Chan et al. [29] also indicated that most past research studies relevant to the use of virtual and remote laboratories focused on engineering sciences, physics, and biology disciplines in general. Nevertheless, there is a need for research specifically focused on the impact of these tools on the quality of sustainability education.

To summarize, this paper is divided into five main sections. In Section 1, we have provided the research context, objectives, and setting. Section 2 summarises the primary trend of studies and relevant case studies in the literature review. Then, a comprehensive

analysis of the remote and virtual laboratory is presented in Section 3. Finally, the discussion and conclusion are given in Sections 4 and 5, respectively.

## 2. Literature Review

The literature review is divided into three key parts. First, an overview of virtual and remote laboratory studies is provided. Second, the trends of the virtual and remote laboratories of engineering education are comparatively analysed. Third, the related case studies during the pandemic are summarised. The literature review is generally based on the journal articles searched on common academic platforms, including Scopus and Web of Science. They are the two most comprehensive multidisciplinary search platforms for published academic research articles.

### 2.1. Overview of Virtual and Remote Laboratories

Experimentation in laboratories is critical in engineering, technology, and science education. Typically, the experiments are conducted in classical experiential labs, which include purchasing of equipment, human resources, and physical presence to sustain them. Accelerating technological advancements and the presence of the internet evolved lab experimentation in technology, engineering, and science, encouraging e-learning or distance learning via virtual and remote laboratories. Heradio et al. [30] addressed that virtual and remote laboratories foster learning outcomes comparable to classical experiential labs. Also, Hmelo et al. [31] identified that virtual and remote laboratories generate remarkable benefits over classical experiential labs. For example, virtual and remote laboratories can be accessible 24/7. However, classical experiential labs are generally accessible for short periods of time because of economic and logistical rationalisation. Realising and using scientific ideas is the outcome of a recurrent learning process that needs experimenting frequently in the lab. As such, classical experiential labs are occasionally inadequate to achieve the desirable influence on students' learning. Raman et al. [28] also reinforced that virtual and remote laboratories have gained comparative advantages over classical experiential labs due to safe learning environments, higher accessibility, decreased cost, adaptability, and timesaving for self-regulated learning. Furthermore, virtual and remote laboratories foster an inclusive, sustainable, and resilient method of supporting training resources, knowledge, and general weaknesses in laboratory skill training.

In particular, virtual and remote laboratories facilitate experimentation related to unobservable circumstances, for instance, electricity, chemical reactions, and thermodynamics. Also, they may fit in reality. In other words, the attributes of the virtual lab's fundamental mathematical model may transform to create more straightforward elaboration of specific circumstances. Additionally, experiments could highlight remarkable information or minimise ambiguous details. The virtual and remote laboratories are a desirable tool to foster pre-laboratory development, which is crucial to enhance the students' lab learning experience [32,33].

### 2.2. Trend of Studies

The literature review conducted in this study aimed to analyse the trends in remote laboratories and virtual laboratories within the context of engineering education. A well-defined methodology guided the review, encompassing the search strategy and transparent criteria for article selection. Two prominent academic databases, Scopus and Web of Science, were employed for data collection. Four specific search strings—"Remote laboratory engineering education" and "Virtual laboratory engineering education"—were used to ensure comprehensive coverage of the relevant literature. Inclusion criteria were established to ensure that selected articles aligned with the review's objectives, including relevance to engineering education, publication in academic journals or conference proceedings, use of the English language, and publication within the timeline from each database's inception to 2021. Conversely, exclusion criteria aimed to maintain the quality and relevance of the selected articles by excluding content unrelated to engineering education, non-English

language publications, and articles published outside the specified timeframe. The collected data were comprehensively analysed to identify trends in publication dates, the trajectory of article numbers over time, and any discernible shifts in research focus. This methodological framework underpinned the review, aiming to provide comprehensive insights into the evolution of remote and virtual laboratories as vital components of engineering education.

The provided data offers insights into the publication history of articles related to remote laboratories and virtual laboratories on Scopus (Figure 1) and Web of Science (Figure 2) databases, spanning from the early 1990s to September 2023. The data reveal that remote and virtual laboratories gained scholarly attention and began to be documented in the literature during the early to mid-1990s. The first remote paper on Scopus was published in 1992, while the first virtual paper was published in 1996. This suggests the emergence of these technologies as research subjects during this period.

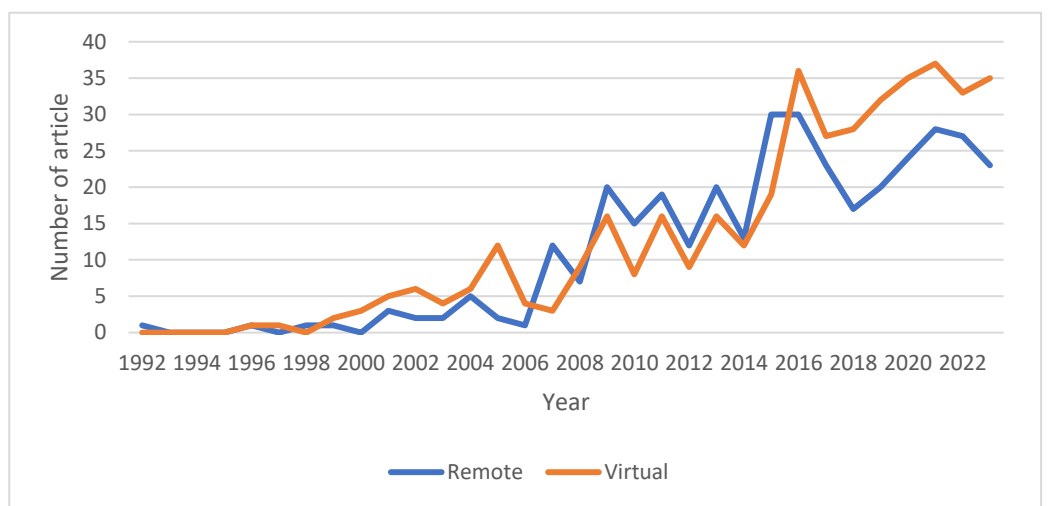

**Figure 1.** Distribution of articles by publishing year on Scopus.

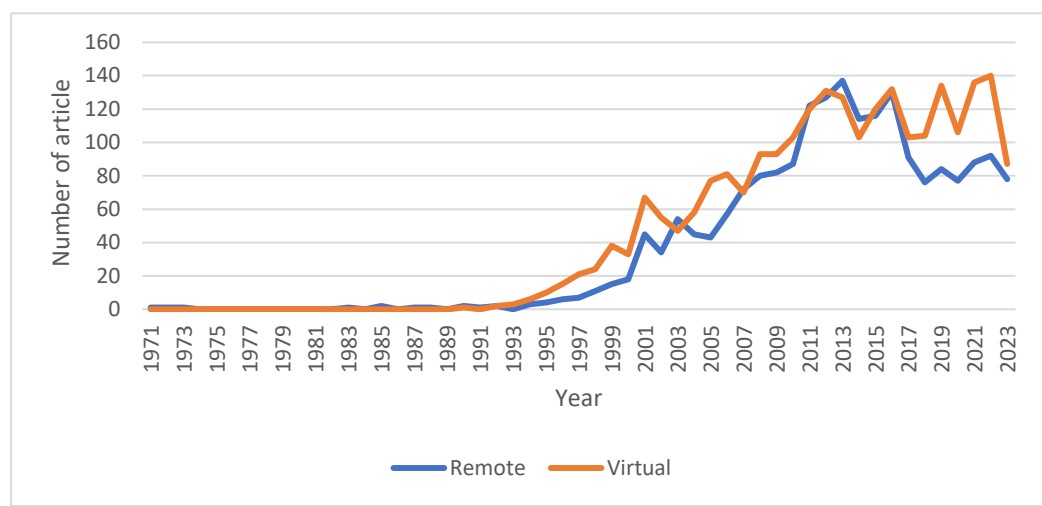

**Figure 2.** Distribution of articles by publishing year on Web of Science.

Regarding the publication trends over time, remote laboratories showed a gradual increase in articles, with a total of 359 articles by September 2023. This publication growth started in 1996 and continued to fluctuate, reaching its peak around 2020. It indicates a growing interest in remote laboratory research over the years. Similarly, virtual laboratories also witnessed an increasing number of articles, totalling 415 by September 2023. Virtual laboratories gained scholarly attention in the late 1990s and continued to attract research interest until 2023. Virtual laboratories consistently attracted more scholarly attention

than remote laboratories throughout this period. Comparing the two types of laboratories from 1996 to 2006, virtual laboratories drew more scholarly attention compared to remote laboratories. This insight could indicate virtual laboratory technology being perceived as more innovative this decade. However, from 2007 to 2016 both remote and virtual laboratories received equal attention, suggesting a balanced focus on both approaches.

After 2016, a notable shift occurred, with virtual laboratories attracting more research studies than remote laboratories. This shift may be attributed to technological advancements, increased interest in online education, and the potential for virtual laboratories to offer diverse learning experiences. In 2020, the COVID-19 pandemic disrupted education systems worldwide, necessitating a rapid transition to online and distance learning. Consequently, there was a significant increase in case studies related to distance learning during the pandemic, underscoring the relevance and importance of remote and virtual laboratories in the context of remote education and their adaptability during crises.

The data were also compared with Web of Science for remote and virtual laboratories. While the Web of Science data started earlier in 1971, it exhibits a similar trend of increasing publications for remote and virtual laboratories over time. Notably, the number of virtual laboratory publications on Web of Science is substantially higher than on Scopus. Overall, the data highlight the evolution of remote and virtual laboratories in academic research, their growing significance in various educational contexts, and their ability to adapt to the changing demands of education, particularly in response to the COVID-19 pandemic.

### 2.3. Case Studies during the Pandemic

Most of the case studies are presented by implementing distance learning in engineering courses. Apart from that, Cheu and Rincones [34] have summarised the findings of a survey conducted about the impacts of COVID-19 on Transportation Engineering Education and Research. Hosting online laboratory sessions is more challenging than hosting other sessions. Remote and virtual laboratories are used as alternative laboratories in online learning to fill the theory-to-practice hole [35]. Allen and Barker [36] have implemented an online gamified virtual lab simulation by Labster [37] to enhance the student experience and create a more engaging and effective learning environment instead of in-person instruction for biomedical engineering courses. Multidisciplinary Engineering Education, a specialist department at the University of Sheffield, delivered the laboratory virtually by providing recorded videos for demonstration and LabVIEW [38] for implementing experiments and mathematical calculations [39]. Barr and Nabir [40] state that lab exercises went well for software engineering classes as they initially had no physical contact.

### 3. Study Analysis

Several studies compare physical and digital laboratories [41–43]. Therefore, an analysis of the virtual laboratory and remote laboratory is undergone in the coming section. It is based on the definitions and perspectives by Faulconer and Gruss [42], and the analysis is divided into learner outcomes, practical skill development, cost, growth potential, accessibility, student–instructor contact time, and safety.

### 3.1. Learning Outcome

Virtual laboratories and remote laboratories are designed and modelled based on the findings from traditional laboratories. Therefore, it is challenging to provide tangible results with sensory feedback. Remote laboratories can provide more realistic experiences than real hardware, typically at a distant location [44]. On the other hand, virtual laboratories can provide better visual experiences. Also, some of them provide an immersive 3D interactive learning environment, which can increase student engagement and students' conceptual understanding [45].

Both virtual and remote laboratories can facilitate active learning and help students develop a deep understanding of sustainability concepts. In addition, virtual laboratories can provide students with a flexible and self-paced learning experience, allowing them

to experiment with various scenarios and learn from their mistakes. On the other hand, remote laboratories provide students with hands-on experience using actual equipment, which can be particularly valuable for students pursuing careers in sustainability-related fields.

### 3.2. Practical Skill Development and Growth Potential

Remote laboratories offer a hands-on experience that allows students to develop practical laboratory skills, such as equipment operation, experimental design, and data analysis. Virtual laboratories can also help students develop these skills, but to a lesser extent. However, virtual laboratories offer a safe and controlled environment, allowing students to experiment with sustainable practices without risk to themselves or the environment. In addition, virtual laboratories offer excellent growth potential, as they can be easily scaled to accommodate more significant numbers of students. On the other hand, remote laboratories may be limited by the availability of laboratory equipment and the number of trained instructors available to support students. Even though this approach works well for knowledge and provides stable learning outcomes, it is limited to developing practical laboratory skills. For example, non-typical laboratories do not require setting up laboratory equipment and cooperating with others. Therefore, they do not have good chances for practical skill development.

### 3.3. Cost

The cost of virtual and remote laboratories can vary depending on a variety of factors, including the type of equipment and software used, the level of complexity of the experiments, and the number of students using the laboratory.

Virtual and remote laboratories generally require less initial investment in equipment and infrastructure than physical laboratories [46]. This is because virtual laboratories can be set up relatively quickly using pre-built software packages or customising existing simulation tools. On the other hand, remote laboratories may require more specialised hardware and software, such as robotic arms or sensors, to enable remote control of experiments. This can result in higher initial setup costs than virtual laboratories.

Virtual and remote laboratories require ongoing maintenance and support, which can add to the overall cost [47]. In the case of virtual laboratories, software updates and patches may be required to ensure laboratories remain up-to-date and secure. Remote laboratories may require more frequent maintenance due to the physical hardware used. Support for both types of laboratories is essential, as students may encounter technical issues or have questions about the experiments. Laboratory developers or instructors can provide this support.

Therefore, virtual and remote laboratories have the potential to be more cost-effective compared to physical laboratories, as they do not require as much initial investment in equipment and infrastructure. However, ongoing maintenance and support costs should also be considered when comparing. Additionally, the cost-effectiveness of virtual and remote laboratories may depend on the specific educational goals and the resources available to the institution or program.

### 3.4. Accessibility

Accessibility is crucial in ensuring that sustainable education is available to all students. Virtual and remote laboratories can be accessed from anywhere in the world, which can make them more accessible for students who are studying abroad or who may not have access to a physical laboratory in their local area. However, due to the limited number of machines and human resources, remote laboratories provide less accessibility than virtual laboratories. It may be relatively less accessible as the non-distant laboratories may have downtime, and the technicians cannot work inside the laboratories [48].

Overall, virtual laboratories and remote laboratories have the potential to be more accessible for sustainable education compared to physical laboratories. This is because they

can offer greater flexibility in terms of time and location and be more effective in terms of time cost. However, it is essential to ensure that virtual and remote laboratories are designed to be accessible to all students, regardless of their abilities or backgrounds. This may require additional considerations, such as providing assistive technologies or ensuring the laboratories are available in multiple languages.

### 3.5. Student–Instructor Contact Time

Instructions can be provided in the laboratory or remotely in other places [49]. Therefore, the amount of student–instructor contact time can vary between virtual and remote laboratories, and both formats have advantages and disadvantages.

In a virtual laboratory, the instructor may have less direct contact with students but may be able to offer more opportunities for asynchronous communication, student autonomy, and collaboration. Because of social distancing, there is less contact time between students and instructors. Virtual laboratories may offer less direct student–instructor contact, as the instructor may not be present in the laboratory with the students. Instead, the instructor may be available remotely through online chat, email, or other communication channels. However, virtual laboratories can offer more opportunities for asynchronous student–instructor contact, as students can communicate with the instructor outside of scheduled laboratory sessions.

Remote laboratories generally offer more opportunities for direct student–instructor contact, as the instructor can be in the remote laboratory during the laboratory session. The instructor can observe and guide students in real-time, answer questions, and provide immediate feedback. Students can also ask questions and seek assistance from the instructor directly, and the instructor can monitor student progress and provide individualised support. In a remote laboratory, the instructor can provide more direct support and feedback but may have less flexibility regarding student autonomy and collaboration. Nevertheless, remote laboratories require instructors, technicians, or senior students.

### 3.6. Safety

When it comes to safety, virtual and remote laboratories offer some advantages over traditional physical laboratories. In virtual laboratories, students conduct experiments using simulated environments and computer software, eliminating the risk of physical injury and exposure to hazardous materials. On the other hand, remote laboratories allow students to control actual equipment in a physical laboratory, but they are not physically present. This can reduce the risk of accidents, as students are not directly exposed to hazardous materials and do not need to handle equipment in the laboratory.

Both laboratory types can provide an extra safety environment for students to implement experiments in front of the computer rather than undergo the experiment in the laboratory, exposing the potential risk to students [50]. However, some safety concerns still need to be addressed in virtual and remote laboratories. Software malfunctions or errors in virtual laboratories could result in inaccurate results or unexpected behaviour that could cause harm. In remote laboratories, there is a risk of equipment malfunction or failure, which could potentially cause harm to the user or damage to the equipment. Additionally, students may not have access to safety equipment or protocols when using remote laboratories, which could increase the risk of accidents.

Ensuring that virtual and remote laboratories are designed and implemented with safety in mind is essential to mitigate these risks. Laboratory developers and instructors should provide students with clear instructions and safety protocols and access to safety equipment and support in case of emergencies. Additionally, it may be necessary to monitor remote laboratories by laboratory staff to ensure that students follow proper safety procedures.

### 3.7. Comparative Finding

Virtual laboratories have several positive aspects. They provide students with an immersive 3D interactive learning environment, fostering engagement and interactivity. Additionally, they offer flexibility in learning, allowing students to progress at their own pace and facilitating active learning through interactive simulations. However, virtual laboratories may struggle to provide tangible results with sensory feedback, limiting the depth of the hands-on experience.

On the other hand, remote laboratories offer distinct advantages. They bring realistic experiences to students located at a distance, enabling them to work with actual laboratory equipment enhancing their practical skills. Similar to virtual laboratories, they also promote active learning. However, remote laboratories have higher initial setup costs and ongoing maintenance and support requirements, which may pose financial challenges.

Regarding learner outcomes, both virtual and remote laboratories promote active learning. Virtual laboratories excel in creating immersive 3D environments and flexibility, while remote laboratories provide hands-on experience with actual equipment, though they may face limitations in creating visual experiences.

Practical skill development and growth potential vary between the two. Virtual laboratories can accommodate more students and offer a controlled, safe environment. In contrast, remote laboratories offer more hands-on experience but might have limitations in developing specific practical laboratory skills.

Considering costs, virtual laboratories require less initial investment in equipment and infrastructure, but they come with ongoing maintenance and support expenses. Remote laboratories, conversely, demand a higher initial setup cost but might be more cost-effective in the long run.

Accessibility is a strong point for both laboratories, as they provide flexibility regarding time and location, cater to multiple languages, and incorporate assistive technologies. However, remote laboratories may encounter accessibility challenges due to downtime and technical constraints.

Regarding student–instructor contact time, virtual laboratories offer more opportunities for asynchronous communication, student autonomy, and collaboration. In contrast, remote laboratories offer more direct student–instructor contact but may require additional personnel, such as instructors, technicians, or senior students.

Lastly, in terms of safety, both virtual and remote laboratories eliminate the risk of physical injury and exposure to hazardous materials, ensuring a safe learning environment. However, they carry the risk of software malfunctions that could lead to inaccurate results or unexpected behaviour, posing potential safety concerns.

## 4. Discussion

Virtual and remote laboratories offer numerous advantages for sustainability education, such as increased accessibility to specialised equipment and flexible scheduling [51]. However, they may not provide the same hands-on experience or access to advanced equipment as traditional labs. Additionally, technical difficulties can disrupt the learning process, and limited collaboration opportunities can hinder teamwork skills. Instructors must also address challenges in preventing cheating, as virtual and remote labs can be accessed from anywhere.

Virtual and remote laboratories can be accessed from anywhere, making it more difficult for instructors to monitor and prevent cheating. Sustainability education often requires a high level of integrity and ethics, and cheating can undermine the integrity of the learning experience. In addition, some students may be tempted to look up answers online or work with others to complete their experiments, which can compromise the learning outcomes of sustainability education.

Sustainability education can be effectively delivered through virtual and remote laboratories, each with unique benefits and limitations. Virtual laboratories are computer-based simulations that allow students to interact with and explore scientific phenomena in

a controlled and safe environment. Virtual laboratories can teach complex sustainability concepts like environmental monitoring, resource management, and renewable energy systems. Students can use virtual laboratory simulations to manipulate variables, collect and analyse data, and draw conclusions, all while minimising the environmental impact of laboratory experiments. Many students can also access virtual laboratories from remote locations, reducing travel and accommodation costs. Remote laboratories, on the other hand, allow students to control real laboratory equipment and perform experiments from a distance. Remote laboratories can be accessed through the internet, allowing students to remotely access and operate real laboratory equipment in a safe and controlled environment. This type of laboratory can teach complex sustainability concepts, such as renewable energy, water treatment, and waste management. Remote laboratories allow students to gain hands-on experience with real equipment, even far from the laboratory. Additionally, remote laboratories can train students in remote or underserved areas where physical laboratories may not be accessible.

There is no better option between virtual and remote laboratories for enhancing engineering education quality as they have pros and cons. Furthermore, some laboratories cannot provide two options, especially during the period with limitations on social contacts. For example, virtual laboratories can provide resources for standard experiments. However, they require sufficient interest before investing in developing a virtual laboratory, and therefore, the recent virtual laboratories are relatively common in colleges and universities. The accessibility of remote laboratories can be enhanced by new technologies, such as speech recognition technology [52]. Also, enhancing internet hardware is beneficial [53]. Furthermore, distance learning requires more instructors to guide the students [54]. Therefore, providing access to academic resources, heartful support, growing trust, and teaching self-advocacy is essential during this difficult period [55].

## 5. Conclusions

In conclusion, virtual and remote laboratories present a nuanced landscape of advantages and disadvantages for sustainable education in engineering. Virtual laboratories excel in providing accessibility, convenience, and cost-effectiveness, making them valuable tools in remote learning environments. However, they may lack the tactile experiences and opportunities for practical skill development offered by physical labs. Conversely, remote laboratories allow students to manipulate actual equipment, fostering hands-on experience even when physically distant. However, they come with challenges related to initial setup costs, accessibility, student–instructor interaction, and safety concerns.

It is essential to recognise that both virtual and remote laboratories have played pivotal roles in addressing the challenges posed by the COVID-19 pandemic, transforming traditional learning methods to cater to the needs of new generations. These laboratories have contributed to enhancing students' well-being in engineering education by enabling technology-enabled socialisation and breaking the confines of traditional classrooms.

Looking forward, continuous assessment of the strengths and weaknesses of virtual and remote laboratories is paramount to ensure their alignment with the evolving needs of students and educators. This research is a substantial contribution to the field of engineering education, with relevance for educators, researchers, policymakers, and learners, especially in the context of international crises such as the COVID-19 pandemic. To build resilient engineering education systems, it is crucial to remain prepared for the future while learning from the experiences of the pandemic.

One critical avenue for future research is the exploration of the effectiveness of virtual and remote laboratories in instilling sustainable practices and attitudes. This includes evaluating their impact on student attitudes towards sustainability and their ability to apply sustainability principles in practical scenarios. Additionally, there is an exciting prospect in integrating virtual and remote laboratories with physical laboratories. This integration could create hybrid learning environments that leverage the advantages of both approaches, providing students with a comprehensive and engaging laboratory experience.

In summary, the landscape of virtual and remote laboratories is dynamic and multifaceted, offering valuable pedagogical solutions while presenting unique challenges. The future of engineering education may well be shaped by the continued evolution and integration of these laboratory methods, contributing to a more flexible, accessible, and effective educational experience for students in the years to come.

**Author Contributions:** Conceptualization, M.C.-P.P. and Y.-y.L.; methodology, M.C.-P.P.; validation, M.C.-P.P., Y.-y.L. and Q.C.; formal analysis, M.C.-P.P.; writing—original draft preparation, M.C.-P.P. and Y.-y.L.; writing—review and editing, M.C.-P.P., Y.-y.L. and Q.C. All authors have read and agreed to the published version of the manuscript.

**Funding:** This research received no external funding.

**Conflicts of Interest:** The authors declare no conflict of interest.

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
