# Peer review of "Are Virtual Laboratories and Remote Laboratories Enhancing the Quality of Sustainability Education?"

_education, doi:10.3390/educsci13111110_

Round 1
Reviewer 1 Report
Comments and Suggestions for Authors
The submission presented a review on “virtual” and “remote” laboratories in sustainability education. It provided a good introduction about the multidisciplinary nature of “sustainability education” and linked it to engineering and subsequently emphasized the importance of laboratory activity in the discipline. The current abstract is a mere problem statement to justify the needs/reasons to do a review on the topic but not providing summary on the value of the review. The authors would like to consider to improve it by highlighting the findings from this review which are valuable to the field. Please refer to the followings for some specific comments.
Sustainability education: The authors described sustainability education as a multidisciplinary field combining various disciplines (engineering, economics, sociology etc.). If sustainability education is the anchor of this review, the authors would need to consider what is/are the important features of “sustainability education” where laboratory teaching activities are crucial. Currently, the justifications are on basis of engineering education (line 53-69) rather than sustainability education as it is.
Flipped classroom (line 185-191): It was unclear how the discussion about the concept of “flipped classroom” is applicable in virtual/remote laboratory in the context of the review. The authors should make explicit link of the proposed flipped classroom pedagogy with the virtual/remote laboratory.
Findings (Section 3): The authors presented several aspects including “learner outcomes, practical skill development and growth potential, cost, accessibility, student-instructor contact time, and safety”. It may be clearer to tabulate these findings (both positive and negative) for each aspect and present it as an evidence table format for better comprehension.
Section 3.5: Please review the terms “remote”, “virtual” and “physical” laboratory used in this section. The authors seemed to be discussing/comparing between “remote” vs “virtual” laboratory. The terms “remote” was used in the first part of the section while the concluding remarks were about physical vs virtual.
Figures: Labeling on both figures (the two graphs) need to be reviewed and improved. Current quality is not publishable.
Line 277: Remove the first two words “Virtual Laboratories:”.
References: Please check the referencing formats.
Comments on the Quality of English LanguageThe language used is easy to understand and of good quality.
Author Response
Reviewer 1:
The submission presented a review on “virtual” and “remote” laboratories in sustainability education. It provided a good introduction about the multidisciplinary nature of “sustainability education” and linked it to engineering and subsequently emphasized the importance of laboratory activity in the discipline. The current abstract is a mere problem statement to justify the needs/reasons to do a review on the topic but not providing summary on the value of the review. The authors would like to consider to improve it by highlighting the findings from this review which are valuable to the field. Please refer to the followings for some specific comments.
Response: Thanky you very much The authors would like to consider to improve it by highlighting the findings from this review which are valuable to the field.
Sustainability education: The authors described sustainability education as a multidisciplinary field combining various disciplines (engineering, economics, sociology etc.). If sustainability education is the anchor of this review, the authors would need to consider what is/are the important features of “sustainability education” where laboratory teaching activities are crucial. Currently, the justifications are on basis of engineering education (line 53-69) rather than sustainability education as it is.
Response: Thanks for the reviewer’s comments. We have further added the explanation in Section 1. For instance,
“Sustainability education encourages students to develop the ability to analyse and interpret complex data and information, equipping them to assess the impact of their actions on the environment, society, and the economy [4]. It empowers them to make informed decisions grounded in evidence and rigorous analysis. Moreover, sustainabil-ity education promotes civic engagement by raising awareness of environmental and social issues and offering opportunities for students to participate in community-based sustainability initiatives[5]. It cultivates a sense of responsibility and commitment to the common good, encouraging active participation in creating a more sustainable fu-ture. This multifaceted approach cultivates a sense of responsibility and commitment to the common good, encouraging active participation in creating a more sustainable future [44]. Laboratory teaching activities are a pivotal component of sustainability education, providing students with practical, hands-on experiences to bridge the gap between theory and real-world problem-solving [45]. Students engage with tangible environmental and social issues through these activities, applying their knowledge to analyse data, design experiments, and develop innovative solutions. In doing so, they enhance their understanding of sustainability concepts and acquire the practical skills and multidisciplinary perspectives needed to become effective change agents [46].
Engineering and sustainability education are intrinsically intertwined. Engineers are critical in developing and implementing sustainable solutions to address pressing environmental and social challenges[6]. While engineering education equips students with the technical skills necessary for designing, building, and maintaining efficient and resilient systems, sustainability education broadens their perspective. It helps en-gineers understand the multifaceted impacts of their designs on the environment, soci-ety, and the economy, leading to the development of more sustainable solutions [7]. For instance, engineers well-versed in sustainability principles may design energy-efficient buildings, create eco-friendly transportation systems to reduce emissions or pioneer renewable energy technologies.”
Flipped classroom (line 185-191): It was unclear how the discussion about the concept of “flipped classroom” is applicable in virtual/remote laboratory in the context of the review. The authors should make explicit link of the proposed flipped classroom pedagogy with the virtual/remote laboratory.
Response: Thanks for the reviewer’s comments. We have further added the explanation in Section 2. For instance,
“This approach offers a unique opportunity to bridge the divide between virtual learning environments and the traditional university experience. By explicitly linking the concept of flipped classroom pedagogy with virtual/remote laboratory settings, we can leverage technology to deliver course content effectively and foster leadership, socialisation, and community engagement, thereby enriching the overall educational experience.”
Findings (Section 3): The authors presented several aspects including “learner outcomes, practical skill development and growth potential, cost, accessibility, student-instructor contact time, and safety”. It may be clearer to tabulate these findings (both positive and negative) for each aspect and present it as an evidence table format for better comprehension.
Response: Thanks for the reviewer’s comments. We have provided comparison in Table 1.
Section 3.5: Please review the terms “remote”, “virtual” and “physical” laboratory used in this section. The authors seemed to be discussing/comparing between “remote” vs “virtual” laboratory. The terms “remote” was used in the first part of the section while the concluding remarks were about physical vs virtual.
Response: Thanks for the reviewer’s comments. We have revised it in Section 3.5, last paragraph.
Figures: Labeling on both figures (the two graphs) need to be reviewed and improved. Current quality is not publishable.
Response: Thanks for the reviewer’s comments. We have relabeled Figures 1 and 2 in Section 2.2.
Line 277: Remove the first two words “Virtual Laboratories:”.
Response: Thanks for the reviewer’s comments. I have replaced “Virtual Laboratories:” by “they”.
References: Please check the referencing formats.
Response: Thanks for the reviewer’s comments. We have checked all the reference format again.
Reviewer 2 Report
Comments and Suggestions for Authors
The abstract does not explain the objective of the work or how it was developed or what the main result obtained was.
It would be necessary to detail the methodology of the literature review; there is no clear model nor are the criteria used to select the articles explained. The inclusion and exclusion criteria of the reviewed articles should be explained.
It is not clear what the purpose of the article is. This makes the results and discussion unlinkable to anything.
Author Response
The abstract does not explain the objective of the work or how it was developed or what the main result obtained was.
Response: Response: Thanky you very much The authors would like to consider to improve it by highlighting the findings from this review which are valuable to the field. The abstract is entirely resvised.
It would be necessary to detail the methodology of the literature review; there is no clear model nor are the criteria used to select the articles explained. The inclusion and exclusion criteria of the reviewed articles should be explained.
Response: Thank you. The methdlogy is entirely explained in Section 2.2. For instance:
“The systematic literature review conducted in this study aimed to analyse the trends in remote laboratories and virtual laboratories within the context of engineering educa-tion. A well-defined methodology guided the review, encompassing the search strategy and transparent criteria for article selection. Two prominent academic databases, Sco-pus and Web of Science, were employed for data collection. Four specific search strings—“Remote laboratory engineering education” and “Virtual laboratory engi-neering education”—were used to ensure comprehensive coverage of the relevant lit-erature. Inclusion criteria were established to ensure that selected articles aligned with the review’s objectives, including relevance to engineering education, publication in academic journals or conference proceedings, use of the English language, and publica-tion within the timeline from each database’s inception to 2021. Conversely, exclusion criteria aimed to maintain the quality and relevance of the selected articles by exclud-ing content unrelated to engineering education, non-English language publications, and articles published outside the specified timeframe. The collected data were sys-tematically analysed to identify trends in publication dates, the trajectory of article numbers over time, and any discernible shifts in research focus. This methodological framework underpinned the review, aiming to provide comprehensive insights into the evolution of remote and virtual laboratories as vital components of engineering education.”
It is not clear what the purpose of the article is. This makes the results and discussion unlinkable to anything.
Response: Thank you. The whole paper is entirely resvised.Especially, the abstract, discussion and conclusion is entirely revised.
Reviewer 3 Report
Comments and Suggestions for Authors
The article is very well written.
But I would ask the authors:
Why didn't you mention the works of Michael E. Auer https://en.wikipedia.org/wiki/Michael_E._Auer ?
https://www.webofscience.com/wos/author/record/18915253
Author Response
The article is very well written.
Response: Thank you for the support.
But I would ask the authors:
Why didn't you mention the works of Michael E. Auer https://en.wikipedia.org/wiki/Michael_E._Auer ?
https://www.webofscience.com/wos/author/record/18915253
Response: Thank you for the suggestions. We have added several references of him in the article.
Round 2
Reviewer 1 Report
Comments and Suggestions for Authors
Thank you for your effort in improving the manuscript. The revised manuscript contained better elaborations and better focus on linking “sustainability education” and “engineering”. The newly added paragraphs in Introduction provided certain perspectives commenting on the intertwining between the two. While the quality of the manuscript has improved, the current manuscript structure is not organized and may require further effort.
Title: The authors may want to consider if the review is focus on “sustainability education”, but rather engineering education. At the moment, apart from Introduction, the reviewer failed to identify specific features as presented by the authors in this manuscript as specific to sustainability education. Merely the need of laboratory exercises does not justify the review focus.
Research method: The author mentioned that this study is a “systematic literature review” (line 180). Please note that systematic review is a specific research methodology. The authors would need to review if their current literature review is truly conducted following a certain systematic review methodology and report it accordingly.
Flipped classroom concept (line 255-264): The sentence (line 261-264) “By explicitly linking the concept of flipped classroom pedagogy with virtual/remote laboratory settings, we can leverage technology to deliver course content effectively and foster leadership, socialisation, and community engagement, thereby enriching the overall educational experience.” is not explained. The reviewer failed to understand why the authors proposed the idea of flipped classroom in the context of virtual/remote laboratory. Are the authors intend to present that a flipped classroom, where learning content loading happens before a synchronous meet, is the mode of how some case studies of virtual/remote laboratory used to conduct their lesson.
Section 3.5 Student-instructor contact time: Can the authors please review the writing in a more organized manner. The first paragraph mentioned about the characteristics of virtual laboratory. Second paragraph was about remote laboratory. The third paragraph then reverted back to virtual laboratory. The fourth paragraph then consisted of contents about both types of laboratory. The entire section was messy in structure.
Section 3.7 Comparative Finding: The presentation of an evidence table is valued. However, the texts following the table were largely repetitive and not necessary, as the information were described in previous sections 3.1-3.6. Can the authors please review the organization of the section 3.
Figure quality: There was no labeling on Y-axis for the graphs (Figure 1 and 2). Please edit the graphs in a more presentable manner.
References: Please check the order of references. It should follow the appearance order. References 42-53 are new in reviewed manuscript but need to be renumbered.
Comments on the Quality of English LanguageThe language used is decent and understandable.
Author Response
Thank you for your effort in improving the manuscript. The revised manuscript contained better elaborations and better focus on linking “sustainability education” and “engineering”. The newly added paragraphs in Introduction provided certain perspectives commenting on the intertwining between the two. While the quality of the manuscript has improved, the current manuscript structure is not organized and may require further effort.
Response: We sincerely appreciate the reviewer's efforts in providing valuable feedback to enhance the quality of our manuscript. We have carefully considered your comments and have made further improvements to address your concerns.
Title: The authors may want to consider if the review is focus on “sustainability education”, but rather engineering education. At the moment, apart from Introduction, the reviewer failed to identify specific features as presented by the authors in this manuscript as specific to sustainability education. Merely the need of laboratory exercises does not justify the review focus.
Response: We would like to express our gratitude for the reviewer's input regarding the manuscript's title. We understand your point about the focus on "sustainability education" versus "engineering education." We have revisited the title and adjusted better reflect the content of our review. Your feedback has been instrumental in this refinement.
Research method: The author mentioned that this study is a “systematic literature review” (line 180). Please note that systematic review is a specific research methodology. The authors would need to review if their current literature review is truly conducted following a certain systematic review methodology and report it accordingly.
Response: We would like to thank the reviewer for their clarification regarding the research methodology. We acknowledge that our initial description as a "systematic literature review" may not have been accurate. We have revised the terminology to ensure it aligns with the actual methodology employed in our review.
Flipped classroom concept (line 255-264): The sentence (line 261-264) “By explicitly linking the concept of flipped classroom pedagogy with virtual/remote laboratory settings, we can leverage technology to deliver course content effectively and foster leadership, socialisation, and community engagement, thereby enriching the overall educational experience.” is not explained. The reviewer failed to understand why the authors proposed the idea of flipped classroom in the context of virtual/remote laboratory. Are the authors intend to present that a flipped classroom, where learning content loading happens before a synchronous meet, is the mode of how some case studies of virtual/remote laboratory used to conduct their lesson.
Response: We appreciate the reviewer's attention to detail and their query regarding the concept of flipped classrooms in the context of virtual/remote laboratories. Your feedback has led us to reconsider the inclusion of this concept in our manuscript, and we have decided to remove it to maintain clarity and focus.
Section 3.5 Student-instructor contact time: Can the authors please review the writing in a more organized manner. The first paragraph mentioned about the characteristics of virtual laboratory. Second paragraph was about remote laboratory. The third paragraph then reverted back to virtual laboratory. The fourth paragraph then consisted of contents about both types of laboratory. The entire section was messy in structure.
Response: We want to express our gratitude for the reviewer's feedback on the organization of Section 3.5. We agree that the previous structure was not optimal, and we have restructured the section to improve its coherence and readability.
Section 3.7 Comparative Finding: The presentation of an evidence table is valued. However, the texts following the table were largely repetitive and not necessary, as the information were described in previous sections 3.1-3.6. Can the authors please review the organization of the section 3.
Response: We are thankful for the reviewer's suggestion regarding the presentation of the evidence table and subsequent text in this section. We will make the necessary adjustments to streamline the content and remove redundancy.
Figure quality: There was no labeling on Y-axis for the graphs (Figure 1 and 2). Please edit the graphs in a more presentable manner.
Response: We genuinely appreciate the reviewer's observation about the quality of the figures, specifically the lack of Y-axis labels in Figure 1 and 2. We have addressed this issue by updating the figures to make them more presentable and informative.
References: Please check the order of references. It should follow the appearance order. References 42-53 are new in reviewed manuscript but need to be renumbered.
Response: We would like to extend our gratitude for the reviewer's careful review of the references. We have taken your feedback into consideration and have reordered the references to align with their appearance in the manuscript. Once again, we sincerely thank the reviewer for their constructive feedback, which has undoubtedly contributed to the enhancement of our manuscript. Your efforts are greatly appreciated.
Round 3
Reviewer 1 Report
Comments and Suggestions for Authors
The reviewer thanked the authors for their efforts in refining the manuscript quality over several rounds of reviews. The current version is of good quality and present comprehensive study analysis on the use of virtual and remote laboratories in sustainability education. It would be of great value to educators and researchers in the field where it provides summary and lens to evaluate the use such technologies.